# Effects of Different Storage Temperatures on Bacterial Communities and Functional Potential in Pork Meat

**DOI:** 10.3390/foods11152307

**Published:** 2022-08-02

**Authors:** Fan Zhao, Zhenqian Wei, Guanghong Zhou, Karsten Kristiansen, Chong Wang

**Affiliations:** 1Laboratory of Genomics and Molecular Biomedicine, Department of Biology, University of Copenhagen, 2100 Copenhagen, Denmark; zhao.fan@bio.ku.dk; 2Key Laboratory of Meat Products Processing, Ministry of Agriculture, Jiangsu Collaborative Innovation Center of Meat Production and Processing, Quality and Safety Control, College of Food Science and Technology, Nanjing Agricultural University, Nanjing 210095, China; 2019108080@njau.edu.cn (Z.W.); guanghong.zhou@hotmail.com (G.Z.); 3BGI-Shenzhen, Shenzhen 518083, China; 4Institute of Metagenomics, Qingdao-Europe Advanced Institute for Life Sciences, BGI-Qingdao, Qingdao 166555, China

**Keywords:** microbial spoilage, pork meat, temperature

## Abstract

Storage temperature is considered one of the most important factors that affect the microbial spoilage of fresh meat. Chilling and superchilling are the most popular storage techniques on the market, but during transportation, the temperature may reach 10 °C and may even reach room temperature during local retail storage. In the present study, we stored fresh pork meat at different temperatures, −2 °C, 4 °C, 10 °C, and 25 °C. The composition and functional potential of fresh or spoiled meat resident microbes were analyzed based on 16S rRNA gene amplicon sequencing. The microbial composition exhibited high similarity between pork meat stored at −2 °C and 4 °C, with *Pseudomonads* and *Brochothrix* being the dominant taxa. *Acinetobacter* sp., *Myroides* sp., and *Kurthia* sp. were markers for spoiled pork meat stored at 25 °C. Both psychrophilic and mesophilic bacteria were observed to grow under a storage temperature of 10 °C, but the overall composition and functional potential based on Kyoto Encyclopedia of Genes and Genomes (KEGG) pathways were found to be similar to that of meat stored at room temperature. Our results broaden the knowledge of possible microbial changes in pork meat during storage, transportation, or retail.

## 1. Introduction

The demand for meat with high nutritional value and delicacy has increased continuously over the last 50 years [1]. In 2016, pork was globally the most consumed meat, accounting for up to 60% of all meat production [2]. However, water and nutrients in pork [3] provide conditions for the growth of microorganisms, such as *Pseudomonas* spp. and *Brochothrix thermosphacta*, which contribute to pork spoilage [4,5]. Thus, pork spoilage caused by the growth and metabolic activities of certain microorganisms has become a major concern in the meat industry [6].

Microbial spoilage of meat can be affected by multiple factors, including meat types, hygiene during slaughter and processing, and storage temperatures. Among these factors, the storage temperature is considered one of the most important factors affecting the growth of bacteria present in meat [4]. Chilling and frozen storage are now the two most commonly used conditions for meat storage [7]. However, recently, there has been increasing interest in applying superchilling for meat storage, whereby the temperature of a food product is lowered to 1–2 °C below the initial freezing point [8], reducing freezing/thawing cycles as well as limiting energy costs by reducing the transport weight, with no need to include ice for cooling as compared to the transport of conventional frozen meat [8]. In addition, the shelf life of chilled storage meat normally ranges from 3 days to 14 days [9,10,11], superchilling can also extend the shelf life by about a factor of 1.5 to 4 compared to chilled storage [12,13]. Although regulation of maximum temperature limits has been established in most countries, it has been reported that temperature may reach 10 °C or even higher during transportation due to an unstable distribution system, transportation distance, and time [4,14]. Such uncontrolled temperature conditions may result in a significantly shortened meat shelf life. Even today, meat in some village markets is still kept in the open air at room temperature with no refrigeration, posing a risk of accelerating meat spoilage [15].

Thus, a better understanding of how microbial composition and functional potential are altered during storage at different temperatures is warranted. Different bacterial populations may contribute to meat spoilage depending on the temperature of storage. During the last decades, high-throughput sequencing and bioinformatics strategies have provided powerful tools for studies focusing on both cultivated and uncultivated bacteria in food, making it possible to comprehensively describe the dynamics of bacterial communities, and to analyze the correlations between microbial metabolic activity and meat spoilage [16,17]. The application of sequencing for characterizing predominant microorganisms involved in the spoilage of pork has been reported previously [2,4,18,19]. In addition, it has been reported that the microbial communities in pork differed substantially upon storage at different temperatures [20,21,22]. However, since most previous studies have focused on the effects of storage at different chilled or superchilling temperatures (including 4 °C, −1 °C, −2 °C, and −3 °C) or different packaging treatments (e.g., modified atmosphere-packing), a comprehensive comparison of changes in the microbial composition and functional potential upon storage temperatures of −2 °C (superchilling), 4 °C (chilling), 10 °C (transportation or retail), and 25 °C (room temperature) is lacking.

In the present study, pork meat was stored at different temperatures, −2 °C, 4 °C, 10 °C, and 25 °C, representing superchilling, chilling, transportation or retail, and room temperature, respectively. The purpose was to evaluate the effects of different storage temperatures on bacterial communities and the functional potentials in fresh and spoiled pork meat. We aimed to provide insight into how different storage temperatures affect resident microbial communities and their possible metabolic activities and, thus, impact the shelf life of pork meat.

## 2. Materials and Methods

### 2.1. Materials and Sample Preparation

A total of 40 *longissimus lumborum* muscles (24 h postmortem) were sampled at Beijing Hualian Group, Nanjing, China, and prepared as previously described [22]. Briefly, all samples were placed in insulated chilled boxes following slaughter and transported to the laboratory within 2 h. Fat and connective tissues were removed. The pretreated muscles were then cut into 5 × 7 × 3 cm chunks and randomly assigned into four groups, representing storage conditions of −2 °C, 4 °C, 10 °C, and 25 °C. The muscles were placed in a plastic tray and then wrapped in commercial polyethylene cling wrap. The wrapped trays were placed in a temperature-stable compressor-cooled incubator (Memmert, Büchenbach, Germany). The status of fresh or spoiled pork meat was monitored according to the measurement of total viable count (TVC), and a TVC value higher than 6 log_10_ CFU/g was considered spoiled. According to previous results we found that spoilage time was 20 h for storage at 25 °C, 72 h for storage at 10 °C, 15 days for storage at 4 °C, and 25 days for storage at −2 °C [23]. Five individual pieces per group representing either fresh or spoiled pork were randomly selected for microbial analysis. Fresh pork samples were collected at the beginning of the experiment as baseline (*n* = 5 per group, 20 in total), and spoiled pork samples were collected at the spoiled time point for each storage temperature (*n* = 5 per group, 20 in total).

### 2.2. DNA Extraction and PCR Amplification

Microbial genomic DNA was extracted from fresh or spoiled pork meat samples using the E.Z.N.A.^®^ Soil DNA Kit (Omega Bio-Tek, Norcross, GA, USA) according to the manufacturer’s instructions. The V4–V5 region of the bacterial 16S rRNA gene was amplified by PCR using the following program: 95 °C for 2 min, 25 cycles at 95 °C for 30 s, 55 °C for 30 s, and 72 °C for 30 s, and a final extension at 72 °C for 5 min. The sequences of the primers used were 515F 5′-barcode-GTGCCAGCMGCCGCGG-3′ and 907R 5′-CCGTCAATTCMTTTRAGTTT-3′. The barcode was an eight nucleotide long sample unique sequence. PCR reactions in a total volume of 20 μL were performed in triplicate. The reaction system contained 10 ng of template DNA, 0.4 μL of FastPfu Polymerase, 4 μL of 5 × FastPfu Buffer, 0.8 μL of each primer (5 μM), and 2 μL of 2.5 mM dNTPs. PCR products were purified from 2% agarose gels using the AxyPrep DNA Gel Extraction Kit (Axygen Biosciences, Union City, CA, USA).

### 2.3. Library Construction and Sequencing

PCR products were quantified by Qubit^®^ 3.0 (Life Invitrogen, Carlsbad, CA, USA) after purification, and amplicons with different barcodes were mixed equally. The pooled DNA product was used for paired-end library construction according to Illumina’s instruction. The amplicon library was sequenced (2 × 250) on an Illumina MiSeq platform by a commercial company (Shanghai BIOZERON Co., Ltd., Shanghai, China). The raw reads were submitted to the NCBI Sequence Read Archive (SRA) database (Accession Number: PRJNA823539).

### 2.4. Processing of Sequencing Data

Raw fastq files were demultiplexed according to the barcode information for each sample using in-house perl scripts with the following criteria: (i) The reads were truncated with an average quality score of less than 20 over a 10 bp sliding window; truncated reads that were less than 50 bp were discarded. (ii) Exact barcode matching, 2 bases of a mismatch for primer matching, and reads containing ambiguous bases were removed. (iii) Only sequences that overlapped with more than 10 bp were assembled. Reads that could not be assembled were discarded.

OTUs were clustered with the threshold of 97% similarity using UPARSE (version 7.1 http://drive5.com/uparse/, accessed on 31 January 2022) and chimeric sequences were identified and removed using UCHIME (version 4.2 https://www.drive5.com/usearch/manual/uchime_algo.html, accessed on 1 January 2022). The phylogenetic annotations of 16S rRNA gene sequences were performed using the uclust algorithm (http://www.drive5.com/usearch/manual/uclust_algo.html, accessed on 1 January 2022) according to the SILVA (SSU138.1) 16S rRNA database with the confidence threshold of 80%.

### 2.5. Alpha-and Beta-Diversity Analyses

The α-diversity (within-sample diversity) was estimated using the Shannon index based on the abundance of reads at the OTU level. The β-diversity analysis between groups was performed and visualized by the principal coordinate analysis (PCoA) using Bray–Curtis dissimilarities based on OTU levels (vegan package, version 2.5–7, R software, Jari Oksanen, Helsinki, Finland).

### 2.6. LEfSe Analysis

A linear discriminant analysis effect size (LEfSe) analysis was performed to identify biomarkers of bacteria in fresh or spoiled meat stored under different temperatures [24]. The Kruskal–Wallis sum-rank test was performed for statistical analysis and a LDA analysis was then applied to determine the effect size of each distinctively abundant taxa (microbiomeMarker package, version 0.99.0, R software, Yang Cao, Tianjin, China).

### 2.7. Prediction of Functional Capacity of the Microbiota

The Phylogenetic Investigation of Communities by Reconstruction of Unobserved States (PICRUSt2) (https://github.com/picrust/picrust2, accessed on 31 March 2022) program based on the Kyoto Encyclopedia of Genes and Genomes (KEGG) database was used to predict the functional potential of the microbiota in samples from the different groups. The visualizations of the data were performed by using ggplot2 (version 3.3.5, R package, Hadley Wickham, Auckland, New Zealand). The PCoA analysis was performed using Bray–Curtis dissimilarities based on the KEGG Orthology (KO) and KEGG pathway (level 2).

### 2.8. Correlation Analysis

Spearman’s correlation coefficients were assessed to determine the relationships between microbial composition and functional potential. The correlation was considered significant when the absolute value of Spearman’s rank correlation coefficient (Spearman’s r) was larger than 0.6 with a *p*-value less than 0.05.

## 3. Results and Discussion

### 3.1. Microbial Composition

Microorganisms are one of the main factors affecting the shelf life of fresh meat [25]. According to our previous study [23], the initial count for TVC ranged between 3.92 log_10_ CFU/g and 4.44 log_10_ CFU/g for the four treatments. Since it has been suggested that meat is spoiled and not suitable for consumption when TVC exceeds 6 log_10_ CFU/g [26], we considered the pork meat spoiled in the current study when TVC reached 7.84 log_10_ CFU/g after 25 days for pork meat stored at −2 °C, 6.21 log_10_ CFU/g after 15 days at 4 °C, 6.94 log_10_ CFU/g after 72 h at 10 °C, and 7.70 log_10_ CFU/g after 15 h at 25 °C, respectively. Consistent with previous studies [1,7], these results show how a lower temperature, as expected, inhibited the growth of the microorganisms, prolonging shelf life before reaching an unacceptable TVC level.

Fresh and spoiled pork meat exposed to different storage temperatures were used for the analysis of the microbial composition. An average of 41,830 high-quality reads per sample with an average length of 422 bp was retained. According to the mapped results, 1554 OTUs were clustered (identity 97%). The alpha diversity of bacterial communities was evaluated by Shannon indices at the OTU level, with the Shannon index increasing with the number of species and the evenness of their abundances in the sample. The Shannon index of spoiled pork meat stored at −2 °C and 4 °C was found to be significantly lower than that of fresh pork meat, while samples stored at 10 °C and 25 °C exhibited significantly higher Shannon indices than that of fresh pork meat (Figure 1A). This indicated that although various types of bacteria may initially contaminate the pork meat [27], the richness and diversity seemed to drop rapidly in response to chilling and supercooling conditions, where only certain types of bacterial species were able to grow during the storage period and further contribute to the spoilage. A higher diversity of bacterial species was found in spoiled pork meat stored at 10 °C and 25 °C compared to storage at lower temperatures. To compare the differences in bacterial composition among samples, PCoA analyses were applied. Although variations in microbial composition were observed in spoiled pork meat stored at different temperatures, the microbial communities in the spoiled pork were all found to be clearly separated from that of fresh pork meat (Figure 1B). Higher similarities in microbial compositions were found between the −2 °C and 4 °C storage groups, and between the 10 °C and 25 °C storage groups.

At the phylum level, six main phyla, including Proteobacteria, Firmicutes, Actinobacteriota, Bacteroidota, Deinococcota, and Myxococcota were identified (Figure 1C). Proteobacteria and Bacteroidota were found to be the dominant phyla in fresh pork meat, followed by Actinobacteriota and Myxococcota. However, in the spoiled pork meat stored at −2 °C and 4 °C, almost only Proteobacteria and Firmicutes were present. For the spoiled pork meat stored at 10 °C and 25 °C, Proteobacteria, Firmicutes, and Bacteroidota were the dominant phyla. In addition, small proportions of Actinobacteriota and Deinococcota were also detectable in the 10 °C storage group.

At the genus level, a total of 378 genera were identified and the top 10 genera are shown in Figure 1D. Various genera were initially present in the fresh pork meat. In the spoiled pork meat stored at −2 °C and 4 °C, *Pseudomonas* and *Brochothrix* accounted for more than 90% of the identified genera resulting in a significant decrease in the diversity compared to fresh pork meat. Similar observations were reported in relation to aerobic packing, where *Pseudomonas* and *Brochothrix* were identified as the predominant bacteria characterizing spoiled meat [4]. Due to the pronounced metabolism of glucose as the first source of energy, *Pseudomonads* may grow faster than *Brochothrix* during the storage period [28]. Our results corroborated this finding as *Pseudomonas* was found to be more abundant than *Brochothrix* in spoiled pork meat stored at 4 °C, and the relative abundance even increased when the storage temperature decreased to −2 °C. For the spoiled pork meat stored at 25 °C, *Acinetobacter*, *Kurthia*, and *Myroides* accounted for about 80% of the genera. Examining the 10 °C storage group, we observed that the dominant genera represented those present both in meat subjected to cooling/supercooling or stored at room temperature, indicating that 10 °C is an intermediate temperature that may allow the growth of both psychrophilic and mesophilic bacteria.

The statistical analysis using linear discriminant analysis effect size (LEfSe) revealed a number of features that enabled discrimination between spoiled pork meat stored at different temperatures and fresh pork meat. *Vibrionimonas* was found to be a marker for fresh pork meat. Currently, relatively little information on *Vibrionimonas* is available. One species belonging to *Vibrionimonas* was reported to be isolated from lake water [29], and another species was identified in tilapia fillet during storage at 0 or −3 °C [30]. Thus, the dominance of *Vibrionimonas* in fresh meat may reflect contamination from the environment during processing. *Pseudomonas* spp. and *Brochothrix thermosphacta* were found to be significantly more abundant in spoiled pork meat stored at −2 °C and 4 °C compared to fresh pork meat (Figure 2A, Padj < 0.05, LDA > 2). *Pseudomonas* spp. are considered the main spoilers of raw meat that has been stored under aerobic refrigerated conditions, and some have been reported to produce volatile metabolites, which are likely derived from the catabolism of amino acids contributing to meat spoilage [31]. *Brochothrix thermosphacta* is another main species responsible for chilled meat spoilage characterized by cheesy or sour odors under aerobic conditions [32]. In the spoilage group stored at 25 °C, *Kurthia* sp. *11kri321*, *Myroides phaeus*, and *Acinetobacter bereziniae* LMG 1003 = CIP70.12 were found to be marker species present in significantly higher abundances than in fresh pork meat. Some species of *Kurthia* are widely distributed in the environment, common in the feces of farm animals, meat, and meat products [33]. They are normally not known to cause spoilage of refrigerated meat, and there is evidence to suggest that their presence is indicative of high temperatures during production or distribution of the product [34]. Our results are consistent with this notion as the relative abundance of *Kurthia* increased in spoiled meat stored at higher temperatures (Figure 1D). *Acinetobacter* spp. and *Myroides* spp. were reported to be dominant at the beginning of chilled pork storage [35]. *Acinetobacter* was shown to be highly abundant in chicken thighs during storage at 10 °C, and *Myroides* increased its abundance to account for 8.1% of all bacterial species [36]. Another study focusing on chicken skin showed that *Acinetobacter* (25.88%) and *Myroides* (13.13%) were the main bacterial species at a storage temperature of 25 °C [37]. Our results differed to a certain extent from these previous reports since a very low abundance of *Acinetobacter* was observed in fresh pork at the beginning of the storage. This may reflect different contamination profiles dependent on the environment and processing. However, after storage at 10 °C and 25 °C, the abundances of these genera tended to be similar to previous findings, pointing to the strong effect of temperature on the microbial composition. The changes of the resident microbiota in pork meat stored at −2 °C or 4 °C have already been extensively studied, but less information is available for storage at 10 °C, and this temperature is actually quite often reached during distribution and in retail cabinets. In this case, temperature fluctuations can significantly affect the growth of microorganisms [38,39]. According to our results, both psychrotrophic and mesophilic bacteria were able to grow during storage at 10 °C, associated with a significantly reduced shelf life as compared to meat stored at −2 °C and 4 °C [40].

### 3.2. Functional Potential of the Bacterial Community

The functional potential of bacterial communities based on KEGG pathways in fresh and spoiled pork meat stored at different temperatures was predicted using PICRUSt2. The PCoA analysis based on the KEGG Orthology and KEGG pathway (level 2) again indicates high similarities between spoiled pork meat stored at −2 °C and 4 °C and between spoiled pork meat stored at 10 °C and 25 °C (Figure 3).

The abundance of pathways at level 2 is shown in Figure 4. Carbohydrate metabolism was the most abundant pathway found in all groups, reflecting that bacteria utilize carbohydrates as the first choice for energy intake until the meat substrate is depleted of glucose and lactate [1]. The mean relative abundance of this pathway was significantly reduced in all spoiled pork meat groups compared to the fresh pork meat group, indicating that during the storage period, carbohydrates were firstly used up and the usage of other nutrients subsequently increased, as exemplified by the significantly higher mean relative abundance of pathways involved in the metabolism of lipids in the spoilage groups compared to fresh meat. Of note, the mean relative abundance of pathways involved in amino acid metabolism was only observed to significantly increase in pork meat stored at −2 °C and 4 °C, while it significantly decreased in meat stored at 10 °C and 25 °C. Microbial amino acid metabolism is closely linked to meat protein degradation during storage, as meat protein can be degraded by both endogenous and microbial proteases into small peptides and amino acids in the middle and late stages of storage [41]. These peptides and amino acids can then be utilized by microorganisms for growth. Our result of the SDS-PAGE analysis indicated a clear association between protein degradation and meat spoilage (Appendix A). More interestingly, the level of TCA-soluble peptides, another index reflecting protein degradation, increased with the storage period in meat stored at −2 °C and 4 °C. However, the level of TCA-soluble peptides first significantly increased from fresh to the semi-fresh stage in meat stored at 10 °C (from 0 to 36 h) and 25 °C (from 0 to 10 h), and then significantly decreased from semi-fresh to spoiled stage (from 36 to 72 h for storage at 10 °C; from 10 to 15 h for storage at 25 °C) (Appendix A). The semi-fresh stage was classified as when the values of pH and TVB-N started to increase significantly according to our previously published data [23]. This result is consistent with the functional analysis of microbial amino acid metabolism, which reflects a higher growth rate and metabolic activity of bacteria at higher storage temperatures depleting the available pool of small peptides and amino acids, as shown by the sharp increase in TVC numbers [23], leading to a bacterial community with a preference for using other nutrients. *Pseudomonas*, a dominating species in meat stored at −2 °C and 4 °C, has been reported to possess high proteolytic activity, capable of switching to using amino acids as the energy source when the substrate is depleted of carbohydrates [1]. This characteristic provides *Pseudomonas* with a competitive advantage over nonproteolytic bacteria [42]. Although *Brochothrix thermosphacta* exhibits no proteolytic activity, it is capable of using lipids when carbohydrates are no longer accessible [43,44]. Spearman’s correlation supported this notion as *Pseudomonas* sp. was positively correlated with amino acid metabolism, and both *Pseudomonas* sp. and *Brochothrix thermosphacta* were positively correlated with lipid metabolism (Figure 5, *p* < 0.01, |r| > 0.6). In addition, we observed a significantly higher mean relative abundance of pathways involved in the metabolism of terpenoids and polyketides in the 10 °C and 25 °C spoilage groups compared to the fresh pork meat group, while the opposite was observed for the −2 °C and 4 °C spoilage groups. Information on the association between meat spoilage and bacterial metabolism of terpenoids and polyketides is limited and further studies are warranted to determine if and to what extent the metabolism of terpenoids and polyketides is relevant in relation to meat spoilage.

## 4. Conclusions

In summary, our study systematically investigated the microbial communities of pork meat stored at different temperatures (−2 °C, superchilling; 4 °C, chilling; 10 °C, transportation or retail; 25 °C, room temperature) using 16S rRNA gene amplicon sequencing. The storage temperature significantly affected the microbial composition in spoiled pork meat. A high similarity was observed between pork meat stored at −2 °C and 4 °C, with *Pseudomonads* and *Brochothrix* being dominant features. *Acinetobacter* sp., *Myroides* sp., and *Kurthia* sp. characterized spoiled pork meat stored at 25 °C. Although, both psychrophilic and mesophilic bacteria may grow under a storage temperature of 10 °C, the overall composition and functional potential of the resident microbiota, especially the functional potential based on the KEGG pathway, were found to show high similarity with meat stored at 25 °C. Our results extend the knowledge of microbial changes in pork meat associated with storage at different temperatures and the relations to meat spoilage, further indicating that the fluctuation of temperature up to 10 °C during transportation markedly affects the composition and functional potential of the microbial communities in pork impinging of meat spoilage. Thus, robust reliable distribution systems considering distance and time are critical to limit meat spoilage.

## Figures and Tables

**Figure 1 foods-11-02307-f001:**
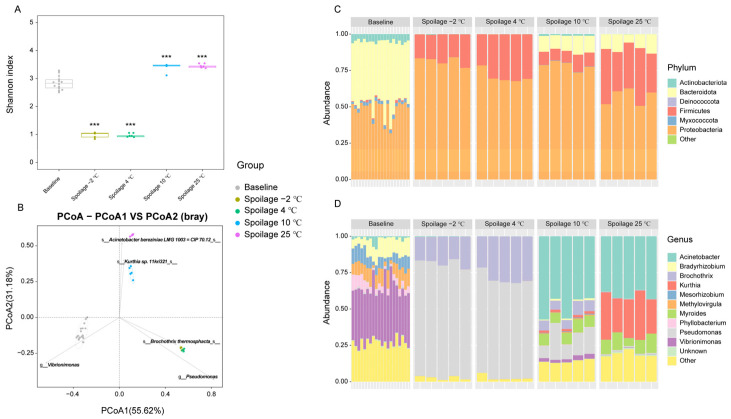
Effect of storage temperature on the microbial diversity and composition. (**A**) α diversity. Shannon index was analyzed at the OTU level. ***: indicates *p* value < 0.001. (**B**) β diversity. The principal coordinate analysis (PCoA) was performed based on the Bray–Curtis dissimilarity at the OTU level. (**C**) Microbial composition at the phylum level. (**D**) Microbial composition at the genus level.

**Figure 2 foods-11-02307-f002:**
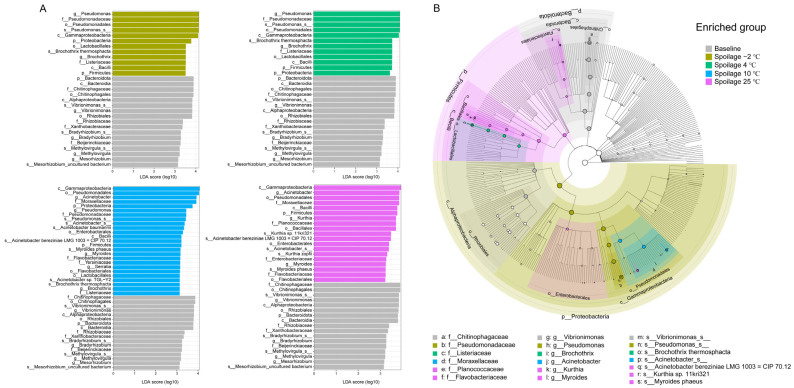
Linear discriminant analysis effect size (LEfSe) analysis of the microbial composition in pork meat. (**A**) Histogram of the LDA scores reveals the most differentially abundant taxa between fresh pork meat and spoiled pork meat stored at each temperature condition, respectively. Only features with least discriminant analysis (LDA) values > 2 and Padj values < 0.05 are displayed. p, phylum; c, class; o, order; f, family; g, genus; s, species. (**B**) Cladogram using the LEfSe method, indicating the phylogenetic distribution of bacteria in fresh and spoiled pork meat.

**Figure 3 foods-11-02307-f003:**
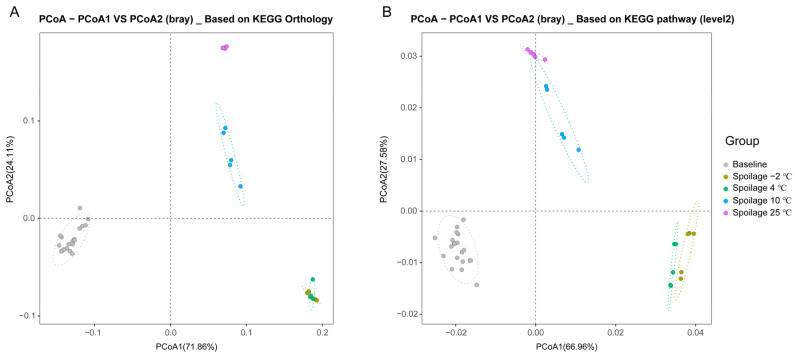
PCoA analysis of the functional potential of microbial communities. (**A**) PCoA analysis using the Bray–Curtis dissimilarity based on the KEGG Orthology. (**B**) PCoA analysis using the Bray–Curtis dissimilarity based on KEGG pathways (level 2).

**Figure 4 foods-11-02307-f004:**
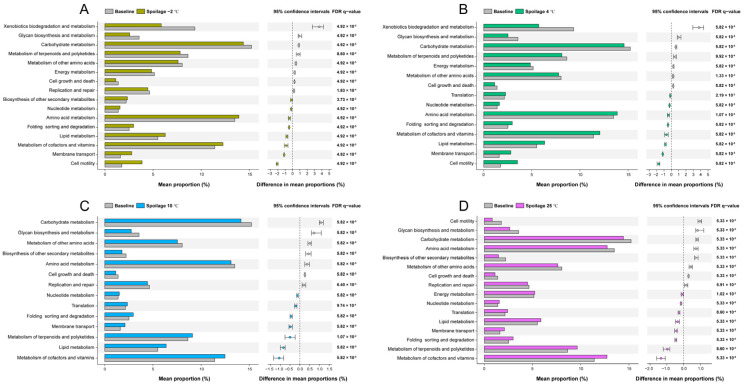
Metabolic pathway profile comparisons between fresh pork meat and spoiled pork meat. (**A**) Comparison between fresh pork meat and spoiled pork meat stored at −2 °C. (**B**) Comparison between fresh pork meat and spoiled pork meat stored at 4 °C. (**C**) Comparison between fresh pork meat and spoiled pork meat stored at 10 °C. (**D**) Comparison between fresh pork meat and spoiled pork meat stored at 25 °C. Pathway comparisons were performed based on KEGG pathways (level 2).

**Figure 5 foods-11-02307-f005:**
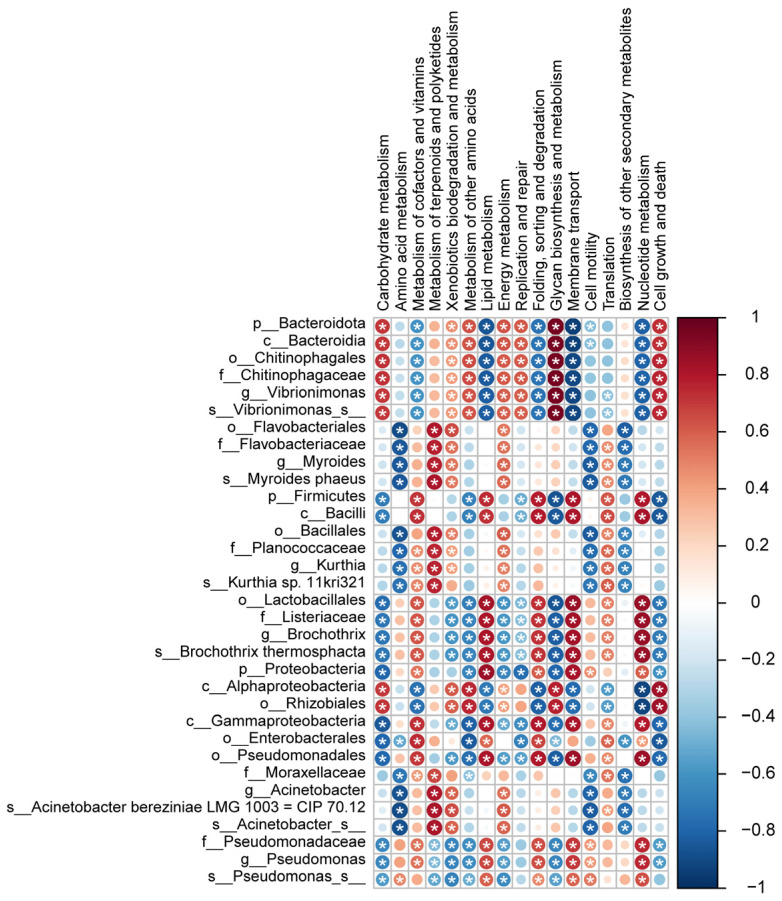
Spearman’s correlation analysis of the association between the microbial community and functional potential. The red color represents a positive correlation while the blue color represents a negative correlation. *: indicates significant correlation (|r| > 0.6, *p* < 0.05).

## Data Availability

The 16S rRNA gene amplicon sequencing data were submitted to the NCBI Sequence Read Archive (SRA) database, accession no. PRJNA823539.

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
