# Peer review of "Effects of Different Storage Temperatures on Bacterial Communities and Functional Potential in Pork Meat"

_foods, 2022, doi:10.3390/foods11152307_

Round 1
Reviewer 1 Report
The submitted article “Effects of different storage temperatures on bacterial communities and functional potential in pork meat” is aimed at evaluating the effect of different storage temperatures on bacterial communities and the functional potentials in fresh and spoiled pork meat. The article is well written and discussed. My comments for further improvement is as follows:
Abstract: Explain in one line or two which cooling technique is used for chilling of meat? Is there any treatment at ambient storage?
Make the colour of text uniform.
Line 49: What is shelf life at chilled storage?
Line 51-52: Explain how this temperature change occurs?
Line 54: “teperature”, write correct spelling.
Line 56-59: Sentence is too lengthy and complex. Rewrite.
Line 65-66: Sentence incomplete.
Need of the study is not justified strongly. Write review of some of the earlier studies and explain knowledge gap from these studies.
Materials and methods: Explain how storage temperature was controlled, whether the meat was packed and if yes, how?
Line 156-158: Though it is considered that meat is spoiled and not suitable for consumption when TVC exceeds 6 log10 CFU/g, why you selected 7.84 log10 CFU/g.
Author Response
Response to Reviewer #1:
Abstract: Explain in one line or two which cooling technique is used for chilling of meat? Is there any treatment at ambient storage?
Response: Due to word limitations, we did not find it possible to include this information in the abstract without losing important information, but this information is already included in the introduction (lines 43-46) and further provided in the materials and method section (line 89). We hope this is acceptable.
Make the colour of text uniform.
Response: We have made the color of text uniform.
Line 49: What is shelf life at chilled storage?
Response: We included this information in the main text, lines 48-49.
Line 51-52: Explain how this temperature change occurs?
Response: We explained that the temperature change was due to an unstable distribution system, transportation distance and time, lines 51-52.
Line 54: “teperature”, write correct spelling.
Response: Sorry for the mistake, we corrected the spelling of the word as temperature, line 56.
Line 56-59: Sentence is too lengthy and complex. Rewrite.
Response: We updated the sentence, lines 57-59.
Line 65-66: Sentence incomplete.
Response: We updated the sentence, lines 67-68.
Need of the study is not justified strongly. Write review of some of the earlier studies and explain knowledge gap from these studies.
Response: We revised the text, lines 68-70.
Materials and methods: Explain how storage temperature was controlled, whether the meat was packed and if yes, how?
Response: We added this information in the main text, lines 88-90.
Line 156-158: Though it is considered that meat is spoiled and not suitable for consumption when TVC exceeds 6 log10 CFU/g, why you selected 7.84 log10 CFU/g.
Response: This threshold was selected based on our previously published paper for the meat stored at -2 ℃. We collected samples every 5 days. On day 20, the TVC was less than 6 log10 CFU/g, on day 25, the TVC was measured as 7.84 log10 CFU/g. Thus, 7.84 log10 CFU/g was chosen as a valid proxy for meat spoilage in the present study.
Reviewer 2 Report
I confirm that authors have expanded introduction and discussion sections and, therefore, tried to reduce plagiarism.
Minor comments:
Although the authors have updated the Introduction section, some references are outdated and could be avoided or updated (6,7,10, 14, 43,44).
Proofread the paper to avoid misspelling such as the one in line 98: "The seuqences".
Author Response
Response to Reviewer #2:
Although the authors have updated the Introduction section, some references are outdated and could be avoided or updated (6,7,10, 14, 43,44).
Response: We have updated the references.
Proofread the paper to avoid misspelling such as the one in line 98: "The seuqences".
Response: Sorry for the mistakes. We have corrected the word in line 105 and also checked for misspellings in the rest of the manuscript.
This manuscript is a resubmission of an earlier submission. The following is a list of the peer review reports and author responses from that submission.
Round 1
Reviewer 1 Report
The research aims to evaluate the microbial spoilage of pork meat in relation to the storage temperatures. The changes and the functional potential of the microbial communities as storage temperature increases from - 2°C to 25°C, also considering the possible environmental effect of intermediate temperature, is interesting and relevant mainly in relation to the global diffusion of this alimentary food. Although the topic is not overly new, to have considered different temperatures, including those hypothetically reached during transport of carcasses or half-carcasses or retail, the paper provides an important contribution to the scientific understanding and discussion of the changes that take place into spoilage of the meat product. The paper is well written, the text is fluent, data elaboration is adequate, and the references, in my opinion, are useful in confirming or discussing the results obtained.
Reviewer 2 Report
The article “Effects of different storage temperatures on bacterial communities and functional potential in pork meat” is well written, with proper methodology and results are very well justified with reasons and similar studies, however, introduction is poorly written. I recommend major revision.
Abstract
Line 17: It should be “in” the market.
Line 25: “KEGG”, write full form at first instance.
Introduction
Line 35: Write the types of microorganisms grow in pork.
Line 42: “limiting energy costs”, How it is reducing energy cost? Mention if you are comparing with anything.
Line 43: “superchilling can also extend the shelf life.” This is a very generalised statement. Compared to what? Mention up to how many days.
Line 45: Provide reason, why temperature may reach 10 °C during transportation?
Line 49-50: “To date, limited information on the changes in the microbial composition under different storage temperatures” – Statement is not correct. There are several research papers available on similar topic. Consequently, the knowledge gap and aim of the research is not clear. Write about the finding of other researchers on refrigerated storage of pork and accordingly rewrite the knowledge gap in this research.
Materials and Methods
Line 75-76: If the research was already conducted by Wei et al 2022 at same storage temperature, why new research was proposed at the same temperature?
Results and Discussion
Line 154: It would be good to first explain what is Shannon indices and its significance in microbial study.
Figure 1 A: Write scale on X axis.
Conclusion
Write about practical applications of knowledge generated from this research.
How the findings of the research will be helpful in averting the increase of temperature up to 10 °C during transportation and increase the shelf life of the pork.
Reviewer 3 Report
In the current investigation, pork meat was kept at various temperatures, including -2°C, 4°C, 10°C, and 25°C. The objective of study was to determine the impact of various storage temperatures on bacterial populations and in fresh and spoiled pork meat. The paper is very simple to consider for this journal and I believe more experiments should be conducted in order to characterize the metabolomic and protein profile. The authors can also provide informations about the shelf life, observing molecules such as biogenic amines.
Apart from lack of information, the article does not provide a sufficient introduction and Introduction and materials and methods sections are plagiarized.
